# Rupestonic Acid Derivative YZH-106 Promotes Lysosomal Degradation of HBV L- and M-HBsAg via Direct Interaction with PreS2 Domain

**DOI:** 10.3390/v16071151

**Published:** 2024-07-17

**Authors:** Lanlan Liu, Haoyu Wang, Lulu Liu, Fang Cheng, Haji Akber Aisa, Changfei Li, Songdong Meng

**Affiliations:** 1Key Laboratory of Pathogenic Microbiology and Immunology, Institute of Microbiology, Chinese Academy of Sciences, Beijing 100101, China; liull925@163.com (L.L.); 18428368009@163.com (H.W.); liull550621@163.com (L.L.);; 2University of Chinese Academy of Sciences, Beijing 100101, China; 3State Key Laboratory Basis of Xinjiang Indigenous Medicinal Plants Resource Utilization, Xinjiang Technical Institute of Physics and Chemistry, Chinese Academy of Sciences, Urumqi 830011, China

**Keywords:** YZH-106, HBV, HBsAg, degradation

## Abstract

Hepatitis B surface antigen (HBsAg) is not only the biomarker of hepatitis B virus (HBV) infection and expression activity in hepatocytes, but it also contributes to viral specific T cell exhaustion and HBV persistent infection. Therefore, anti-HBV therapies targeting HBsAg to achieve HBsAg loss are key approaches for an HBV functional cure. In this study, we found that YZH-106, a rupestonic acid derivative, inhibited HBsAg secretion and viral replication. Further investigation demonstrated that YZH-106 promoted the lysosomal degradation of viral L- and M-HBs proteins. A mechanistic study using Biacore and docking analysis revealed that YZH-106 bound directly to the PreS2 domain of L- and M-HBsAg, thereby blocking their entry into the endoplasmic reticulum (ER) and promoting their degradation in cytoplasm. Our work thereby provides the basis for the design of a novel compound therapy to target HBsAg against HBV infection.

## 1. Introduction

Hepatitis B virus (HBV) infection affects about 250 million patients worldwide and poses a major global health problem. HBV infectious virions have an icosahedral nucleocapsid composed of hepatitis B core protein (HBc), viral polymerase (Pol), and viral genome DNA. The nucleocapsid is surrounded by a viral envelope containing large, middle, and small viral surface antigens (HBs) [1]. Hepatitis B surface antigen (HBsAg) proteins differ in their N-terminus but share a common S domain on their C-terminus. The middle surface antigen (M-HBsAg) contains an additional region, the preS2 region, at the N-terminus of the small surface protein (S-HBsAg), and the large surface antigen (L-HBsAg) carries the PreS2 and PreS1 domains at the N-terminus of the S-HBsAg [2]. Under HBV infection, the HBsAg production and secretory pathway and the viral replication pathway are largely distinctive processes within the hepatocyte [3]. After translation from two HBV sub-genomic mRNA transcripts in cytoplasm, the small, middle, and large surface proteins assemble in the endoplasmic reticulum (ER) to generate noninfectious sub-viral particles (SVPs) with either a spherical or a long filamentous form. The SVPs are secreted via the Golgi pathway or multivesicular body-associated endosomal sorting complex required for the transport (ESCRT) machinery. Meanwhile, viral genome-containing nucleocapsids are assembled with the three surface proteins to form infectious virions, and then are secreted by the ESCRT pathway.

Chronic HBV infection (CHB) is characterized by large amounts of complete virus, as well as noninfectious envelope particles that are secreted in serum at levels far in excess of mature virions and are believed to play a key role as a decoy for antiviral immunity. It has been reported that monocytes, dendritic cells, nature killer cells, and nature killer T cells were inhibited by direct interaction with HBsAg. Large numbers of HBsAg could also cause exhaustion of virus-specific cytotoxic T lymphocytes (CTLs) and helper T (Th) cells. Therefore, HBsAg acts not only as the biomarker of HBV infection and expression activity in hepatocytes, but also majorly contributes to HBV persistent infection [4,5,6].

Recent studies on CHB therapy have been focused on establishing a functional cure, defined as sustainable HBsAg seroclearance (HBsAg loss) and undetectable serum HBV DNA levels, with or without HBsAg antibody seroconversion [7,8,9]. Anti-viral therapies targeting HBsAg include RNA interference (RNAi), anti-sense oligonucleotide (ASO), and nucleic acid polymers, selectively suppressing SVPs assembly and/or secretion [10]. RNAi-based therapy against CHB has been studied in Phase II clinical trials, and it has been demonstrated that HBsAg can be effectively reduced by this treatment, and thus it holds promise for HBsAg loss [11,12].

Artemisia rupestris L. is a traditional herb with antitumor, detoxification, anti-antiviral, antibacterial, and anti-inflammatory activities [13,14]. Rupestonic acid can be extracted from Artemisia rupestris L., and more than 200 rupestonic acid derivatives have been synthesized by researchers. Compound YZH-106, a member of the rupestonic acid derivatives with phenyl isoxazole modified to its carboxyl group, displayed activities against a broad spectrum of influenza viruses (IAV), including drug-resistant IAV strains [15,16]. In this study, we provide experimental evidence that YZH-106 could bind directly to the PreS2 domain of L- and M-HBsAg, which blocked L- and M-HBsAg entry to the ER and promoted their degradation by lysosome. Therefore, our results reveal a novel mechanism by which HBsAg was targeted for degradation. The results may offer a novel therapeutic strategy for HBV infection treatment.

## 2. Material and Methods

### 2.1. Reagents and Antibodies

The following reagents and antibodies were obtained as indicated: YZH106 with more than 98% purity was originally provided by the Xinjiang Technic Institute of Physics and Chemistry, Chinese Academy of Sciences. HBsAg antibody, PreS2 antibody, Calnexin antibody, Lamp1 antibody, and ubiquitin antibody were purchased from Abcam. GAPDH antibody and HRP-conjugated secondary antibody were obtained from Beijing Zhong Shan Golden Bridge Biotechnology. Lamivudine and MG132 were obtained from Selleck Chemicals LLC, and the ECL Plus chemiluminescence system was obtained from Beyotime Biotechnology.

### 2.2. Cell Culture and Transfection

The human hepatoma cell line Huh-7 was obtained from the ATCC (Manassas, VA, USA), the HepG2 cells were stably transfected with two copies of the HBV genome, and the HBV-replicating HepG2.2.15 cells were acquired. The hepG2.2.15 cells were purchased from Cellcook Biotech Co., Ltd. (Guangzhou, China). The cells were cultured in Dulbecco’s modified Eagle medium (DMEM, Gibco, Grand Island, NY, USA) supplemented with 10% FBS (Invitrogen, Carlsbad, USA), 100 μg/mL streptomycin, and 100 IU/mL penicillin at 37 °C in a 5% CO_2_ incubator. For the hepG2.2.1.5 culture, 380 μg/mL G418 (Invitrogen) was added to DMEM. The hepG2-hNTCP cells, which stably express a functional receptor for HBV human sodium taurocholate co-transporting polypeptide (hNTCP) in HepG2 cells, were provided by Prof. Wenhui Li (National Institute of Biological Sciences, Beijing, China). The hepG2-hNTCP cells were maintained in a hepatocyte maintenance medium (PMM) containing Williams E medium supplemented with 5 μg/mL transferrin, 10 ng/mL EGF, 3 μg/mL insulin, 2 mM L-glutamine, 18 μg/mL hydrocortisone, 40 ng/mL dexamethasone, 5 ng/mL sodium selenite, 2% DMSO, 100 U/mL penicillin, and 100 μg/mL streptomycin. The cells were maintained in a 5% CO_2_ humidified incubator at 37 °C, and the medium was changed every 2–3 days.

pCDNA 3.1 containing 1.3 copies of the full-length HBV genomic sequence (D genotype) was constructed in our lab and named pHBV1.3 plasmid. The Huh-7 cells were washed twice with Opti-MEM (Invitrogen) and transfected with pHBV1.3 plasmid using the Lipofectamine 2000 reagent (Invitrogen).

### 2.3. HBV Infection Assay

The HBV genotype B virus was obtained by ultracentrifugation of plasma from three chronic HBV carriers. CHB patients were defined as follows: people who had had chronic HBV infection with HBsAg-positive serum for ≥6 months and may have exhibited symptoms of hepatitis and abnormal hepatic function. Patient sera were concentrated with Ultra-15 centrifugal filter units containing Ultracel-50 membranes (Millipore Corp., Bedford, MA, USA). After cesium chloride density gradient ultracentrifugation and dialysis against PBS buffer, HBV virions were prepared for the infection of HepG2-NTCP cells.

The viral infections were conducted in 12-well plates at multiplicities of genome equivalents of 100. Briefly, 3.2 × 10^5^ epG2-hNTCP cells were inoculated with 3.2 × 10^7^ copies of genome equivalent virus and incubated for 16 h in PMM medium containing 4% PEG-8000. The cells were then washed with medium three times and maintained in PMM medium for 3 d.

### 2.4. Detection of HBsAg and HBeAg

The expression levels of HBsAg and HBeAg in the cell supernatants were determined using enzyme-linked immunosorbent assay (ELISA) kits (Kehua Bio-engineering Co., Ltd., Shanghai, China) according to the manufacturer’s instructions. All experiments were conducted in triplicate.

### 2.5. Cell Proliferation Assay and Cell Viability Assay

Cell proliferation was carried out using the Cell-Counting Kit (CCK)-8 (Dojindo, Kumamoto, Japan), as described in [17]. In brief, 1.0 × 10^4^ cells were plated per well of 96-well plates and incubated overnight. Then, the cells were treated with or without compound at the indicated concentrations. After 24 h, the culture medium was changed with 10% CCK-8 solution. and the cells were incubated for 1 h at 37 °C in a humidified 5% CO_2_ atmosphere. After incubation, absorbance was read at 450 nm for CCK-8 using a spectrophotometer, and the quantity of formazan product was directly proportional to the number of living cells in the culture.

Cell viability was determined using the CellTitre-Glo (CTG) luminescent cell viability assay (Promega, Madison, WI, USA). The cells were seeded in white 96-well plates at a density of 1.0 × 10^4^ cells per well. After incubation overnight, the cells were treated with drug for 72 h. Luminescence was measured using a CLARIOstar Plus (BMG LABTEC, Ortenberg, Germany) and compared to the DMSO-treated cells.

### 2.6. RT-PCR Analysis

Viral DNA copy numbers in the culture medium were detected using an HBV nucleic acid RT-PCR kit (Bioer Technology, Hangzhou, China) at 24 h after the addition of YZH-106. The total RNA was extracted from the cells using TRIzol (Invitrogen). First-strand cDNA was synthesized with an oligo(dT)-adaptor primer and AMV reverse transcriptase (Takara, Tokyo, Japan). The PCR primers are listed in Table 1, with the β-actin gene serving as an internal control. Amplification products were analyzed by 1.5% agarose gel electrophoresis.

### 2.7. Immunoprecipitation (IP) and Western Blot Analysis

After treatment with YZH-106, the cells were harvested and lysed using ice-cold NP40 cell lysis buffer. Equal amounts of total proteins were incubated with 2 µL HBsAg antibody or IgG as a control for 2~3 h at 4 °C. Then, protein A and G Sepharose beads (Santa Cruz Biotech, Santa Cruz, USA) were added and incubated with the cell lysates overnight at 4 °C. The beads were washed four times with cell lysis buffer and resuspended in 5X SDS-PAGE loading buffer. The proteins were separated by SDS-PAGE and analyzed using Western blot.

### 2.8. The Endoplasmic Reticulum and Cytoplasm Components Isolation

Isolation of the ER proteins was performed with an ER protein extraction kit (BB-31454, BestBio, Shanghai, China), and cytoplasmic proteins were extracted from the cells using special lysis buffer (BB-36021, BestBio). In brief, the cells were harvested after centrifugation at 500× *g* for 5 min and washed with cold PBS. For the ER protein extraction, a Dounce homogenizer was used to fully homogenize the cells after Solution A was added. After centrifugation at 1000× *g* for 10 min at 4 °C, the supernatant was collected and centrifuged at 12,000× *g* for another 10 min at 4 °C. The pellet was then resuspended in Solution B and centrifuged at 45,000× *g* for 45 min at 4 °C. Finally, the pellet was resuspended in Solution C, and the ER proteins were acquired. For isolating the cytoplasmic protein, a cold special lysis buffer was added to the cell pellet. After the pellet was mixed by vortex and kept on ice for 30 min, the cell lysate was centrifuged at 1200× *g* for 5 min at 4 °C. The supernatant was collected to obtain the cytosolic fraction.

### 2.9. Immunofluorescence (IF)

After the cells were treated with YZH-106 for 24 h, they were fixed with 4% paraformaldehyde and incubated with 5% BSA. Then, the cells were incubated with primary antibody and TRITC or FITC-conjugated secondary antibody. Images were obtained using a Leica TCS SP8 confocal laser-scanning microscope (Leica Microsystems, Wetzlar, Germany).

### 2.10. Expression and Purification of Protein

The optimized coding sequence of the HBV PreS1-PreS2 (GenBank: CCH63721.1) was cloned into the pET-28a (+)-sumo expression vector, and the resulting vector was named pET-28a-PreS. This vector included an N-terminal 6×His tag and a C-terminal SUMO tag. PreS1-PreS2 protein was expressed in E. coli BL21(DE3) pLysS (Invitrogen, Madison, WI, USA) with induction of IPTG. Bacteria were lysed by sonication, and supernatant was acquired after centrifugation. Protein was then purified sequentially through a His-trap HP column (Cytiva, Marlborough, MA, USA) and a HiTrap Q HP column (Cytiva). The purified protein was assessed using 15% Coomassie-stained SDS-PAGE.

### 2.11. Surface Plasmon Resonance (SPR) Assay

The affinity between PreS1-PreS2 protein and YZH-106 was measured using a Biacore 8K system. PreS1-PreS2 protein was immobilized on CM5 chips (Cytiva) at a concentration of 1 mg/mL, and serial dilutions of YZH-106 were added.

### 2.12. Molecular Docking

The three-dimensional structures of Pre-S1 and PreS1-PreS2 proteins were predicted using AlphaFold following standard protocols. Molecular docking of YZH-106 with these proteins was performed with MOEDOCK (MOE), utilizing the AMBER10: EHT force field and the R-field implicit solvation model for optimization. Ligand structures were obtained from PubChem and prepared using MOE’s energy minimization. Binding sites were identified with MOE’s ‘Site Finder’, focusing on specific residues for each protein. The most likely binding modes were determined based on the lowest binding free energy, visualized using PyMOL.

### 2.13. Ethics Statement

All human subjects provided written informed consent. The study protocol was approved by the Ethics Committee of the Fifth Medical Centre of the PLA General Hospital.

### 2.14. Statistical Analysis

Data were expressed as percentage, mean, ± SD. Comparisons between two groups were analyzed with the Student t test. *p* < 0.05 was considered statistically significant.

## 3. Results

Rupestonic acid derivative YZH-106 suppresses HBsAg secretion

In this study, we first examined whether the rupestonic acid YZH-106 could suppress HBV expression and replication. As seen in Figure 1A,B, HBsAg and HBeAg secretion in HepG2.2.15 cells with stable transfection HBV expression were reduced by YZH-106 treatment in a dose-dependent manner. Moreover, YZH-106 caused more decrease in HBsAg than HBeAg. To determine if the decrease in HBV replication caused by YZH-106 was related to the inhibition of cell proliferation induced by YZH-106, CCK-8 analysis was performed to examine the effect of YZH-106 on the proliferation of HepG2.2.15 cells. As illustrated in Figure 1C, treatment with YZH-106 at concentrations of 4 μM or less had no significant effect on HepG2.2.15 cell proliferation in the CCK8 analysis, excluding the possibility that YZH-106 affected HBV expression through the inhibition of cell growth. Thus, 3 μM YZH-106 was selected to perform the following experiments.

As shown in Figure 1D, treatment with 3 μM YZH-106 caused approximately 58% and 48% reductions in the levels of HBsAg and HBV-DNA in the cell supernatant (both *p* < 0.001). Lamivudine, as the positive control, could only reduce HBV-DNA in the cell supernatant by about 35% (*p* < 0.01). In contrast, no obvious changes in HBV mRNA levels were observed in the YZH-106-treated cells (Figure 1E), indicating that YZH-106 affected HBsAg levels not through the inhibition of HBV transcription, while the levels of HBsAg or HBV mRNA were hardly affected by lamivudine (Figure 1E). Furthermore, the HepG2-NTCP cells infected with HBV from CHB patients were treated with YZH-106, and then HBsAg, HBeAg, and HBV-DNA in the cell supernatant were detected. As shown in Figure 1F, YZH-106 induced a reduction in HBV DNA copies and HBsAg and HBeAg levels in a dose-dependent manner. More decrease in HBsAg than HBeAg was observed after treatment with YZH-106. The EC50s of YZH-106 on HBV-DNA, HBsAg, and HBeAg were around 4 μM, 3 μM, and 8 μM, respectively. To exclude the possibility that inhibition of HBV by YZH-106 was caused by the cellular cytotoxicity in the HepG2-NTCP cells, the cell viability was analyzed using the CellTiter-Glo assay. The results demonstrated that YZH-106 at concentrations of 4 μM or less could hardly influence cell viability (Figure 1G), indicating that YZH-106 exerted antiviral activity not mainly through cytotoxicity.

YZH-106 promotes the lysosomal degradation of L- and M-HBs proteins.

To investigate the underlying mechanism of the YZH-106-mediated inhibition of HBsAg secretion, levels of HBV envelope proteins L/M/S-HBs were determined. Western blot analysis revealed that the L- and M-HBsAg but not the S-HBsAg levels in the HepG2.2.15 cells or Huh-7 cells transfected with pHBV1.3 were reduced under YZH-106 treatment compared to the control (Figure 2A). YZH-106 induced a similar decrease in L- and M-HBs in these two HBV-transfected cell models. In addition, a reduction in L- and M-HBs was also observed in the HBV-infected HepG2-NTCP cells under YZH-106 treatment compared to the control (Figure 2A). As YZH-106 inhibited HBsAg production not via transcriptional regulation, the stability of the HBV envelope proteins was then analyzed. As seen in Figure 2B, YZH-106 caused a dramatic reduction in the protein stability of L- and M-HBs in the presence of the protein synthesis inhibitor cycloheximide (CHX). Compared to the control, no obvious change in the levels of HBs ubiquitination was observed in the YZH-106-treated cells (Figure 2C). In addition, YZH-106-induced reduction of L- and M-HBs could largely be restored by treatment with the lysosomal protease inhibitors E64d and Pepstatin A (Figure 2D). Further, fluorescence microscopy was used to analyze the influence of YZH-106 on the co-localization of PreS2 and lamp1, the marker of lysosome. As illustrated in Figure 2E, co-localization of PreS2 and lamp1 was significantly increased upon treatment with YZH-106. Altogether, these data indicate that YZH-106 mediated downregulation of L- and M-HBs mainly through lysosomal degradation.

The L- and M-HBs levels in the ER but not in the cytoplasm were obviously reduced upon YZH-106 treatment.

After HBs proteins are translated in cytoplasm, they quickly translocate to and accumulate in the ER and form agglomerates through covalent disulfide bridges with different cysteines in their S domain, which are then secreted from the cell as new infectious virions as well as noninfectious envelope particles [17]. We then investigated at which step YZH-106 may affect HBsAg secretion. The cytoplasmic and ER components in the HepG2.2.15 cells under YZH-106 treatment for 48 h were isolated, and the HBs proteins were examined using Western blot analysis. As demonstrated in Figure 3A, treatment with YZH-106 led to decreased L- and M-HBs but not S-HBs protein levels in the ER compared to the control, whereas there was almost no change in the L- and M-HBs protein levels in the cytoplasm upon treatment with YZH-106. The co-localization of PreS2 and the ER marker calnexin or the cytoplasm marker HSP70 was further evaluated using a fluorescence microscope. As shown in Figure 3B,C, co-localization of PreS2 and calnexin but not HSP70 significantly declined with YZH-106. Collectively, these data suggest that YZH-106 suppressed L and M-HBs protein entry into the ER.

YZH-106 directly binds to the PreS2 domain in L- and M-HBs proteins.

To explore the underlying basis for the YZH-106-mediated blockage of L- and M-HBs entry into the ER, we determined to test if YZH-106 could bind to L- or M-HBs proteins. As YZH-106 was shown to induce degradation and block ER entry of L- and M-HBs but not S-HBs proteins, we speculated that YZH-106 may interact with the shared PreS2 domain of L- and M-HBs. His-tagged PreS1-PreS2 fusion protein expressed in *E. coli* was analyzed using Coomassie blue staining and Western blotting (Figure 4A). The interaction of YZH-106 and PreS1-PreS2 protein was evaluated with a direct binding assay using the Biacore system. As illustrated in Figure 4B, YZH-106 bound to a PreS1-PreS2-conjugated sensor chip with fast kinetics. By using steady-state binding analysis, the binding affinity (Kd) was calculated at 279 μM.

Finally, molecular docking was performed with a ZDOCK Server (version 3.0.2) to analyze the intra-molecular interactions between YZH-106 and PreS1-PreS2 or PreS1 protein. PreS1-PreS2 or PreS1 protein structures were firstly predicted using AlphaFold 3 software, then the proteins were separately docked and modeled in a complex with YZH-106. It was found that the potential binding sites for YZH-106 and PreS1-PreS2 protein spanned from Arg102 to Ser152 adjacent or within PreS2, while no binding sites were observed within the PreS1 domain (Figure 4C and Appendix A). Furthermore, an MOE-Dock simulation study was carried out to determine the binding affinity of the compound with proteins. The docking score is summarized in Table 2. The more negative the docking score, the better the binding affinity between the compound with proteins. The docking score between the YZH-106 compound with PreS1 was −2.89 kcal/mol, and the docking score between YZH-106 with PreS1-PreS2 protein was −6.77 kcal/mol, indicating a relatively much stronger affinity between YZH-106 and PreS1-PreS2.Together, these results validate that YZH-106 binds directly to the PreS2 domain of L- and M-HBs proteins, which may inhibit their translocation to the ER.

## 4. Discussion

Chronic HBV infection causes severe liver disease, including cirrhosis and liver failure, and increases the risk of hepatocellular carcinoma occurrence. Current antiviral treatments for CHB include PEG-interferon and nucleos(t)ide analogues (NAs). However, both agents have limited therapeutic efficacy in HBsAg loss [9,18]. Our data demonstrated that a rupestonic acid derivative YZH-106 efficiently suppressed HBsAg secretion in HBV stably expressed HepG2.2.15 cells and HBV-infected HepG2-NTCP cells. Further investigation revealed that the protein levels of the surface proteins L-HBs and M-HBs were notably decreased via a lysosomal degradation pathway under treatment with YZH-106. Biacore and docking analysis revealed that YZH-106 bound directly to the PreS2 domain of L- and M-HBs, thereafter blocking their entry to the ER and leading to their degradation (Figure 5). Our results offer a basis for developing YHZ-106 as a potential antiviral agent for HBV infection.

It has been reported that a high level of HBsAg could lead to a reduction in CD8+ T cell function and T cell exhaustion. In addition, HBsAg promotes the disease progression of CHB to liver cirrhosis and HCC [19,20]. Sustained HBsAg loss is defined as a functional cure of CHB, which is related to improved clinical outcomes. Therefore, one of the primary goal of treatment outcomes for CHB is the seroclearance and conversion of HBsAg [21,22]. Current anti-HBV agents inhibit HBsAg expression and secretion through various mechanisms, such as the suppression of HBV entry or HBsAg release, siRNA, ASO, neutralization of HBsAg release, and inhibition of cccDNA [22,23,24]. It has been reported that siRNA- and ASO-based treatment strategies against CHB led to an HBsAg decrease to very low levels, which resulted in the reversion of T cell anergy and reconstitution of the host immune response [11,25,26]. Up to now, natural compounds that exert their antiviral effects by direct binding to HBV surface proteins have not been reported. In our present study, we demonstrated that the rupestonic acid derivative YZH-106 could bind to the PreS2 domain directly and cause degradation of L- and M-HBs proteins. In general, the correct ER entry of newly synthesized secretory and transmembrane proteins is achieved by two major pathways: signal recognition particle (SRP)-dependent and SRP-independent pathways [27]. The SRP-dependent process involves the recognition of signal peptides, binding of transport complexes, and transport through the ER membrane [28,29,30]. We inferred that the binding of YZH-106 with the PreS2 domain might inhibit the entry of L- and M-HBs proteins into the ER by interrupting the process of protein entry into the ER, thereby resulting in their lysosomal degradation. This is somehow different from the previously reported mechanism of the YZH-106-mediated inhibition of the influenza virus, which revealed that YZH-106 was not likely to directly target viral components, but inhibited IAV replication by activation of an HO-1-mediated type I IFN response [16]. The importance of a stable proportion of L-HBs, M-HBs, and S-HBs in the envelope of infectious as well as noninfectious viral particles is well established. Gerken et al. demonstrated that during the transition from acute hepatitis B to HBsAg loss, the proportion of L-HBs and M-HBs in viral envelope decreases [2,3,4,5,6,7,8,9,10,11,12,13,14,15,16,17,18,19,20,21,22,23,24,25,26,27,28,29,30,31]. The roles of L-HBs in viral entry, morphogenesis, and output are critical. L-HBs-negative mutant strains are unable to form or secrete viral particles, suggesting that a reduction in L-HBs correlates with a decrease in viral replication [32]. Meanwhile, M-HBs have been shown to play a regulatory role in HBV replication, with their absence leading to reduced secretion of viral particles [33]. Pfefferkorn et al. also observed a decrease in L-HBs and M-HBs proportions prior to total HBsAg loss during NA and PEG-IFN treatment [34]. These studies suggest that inhibition of L- and M-HBs may lead to decreased HBV replication and eventually HBsAg loss.

In this study, we observed that a reduction in L- and M-HBs levels induced by YZH-106 was largely restored by lysosomal enzyme inhibitors, indicating that YZH-106 mainly mediated lysosomal degradation of the viral surface proteins.

Protein degradation pathways are classified into proteasomal degradation and lysosomal degradation that can be further divided into three types: macroautophagy, microautophagy, and chaperone-mediated autophagy [35,36]. Interestingly, a previous study showed that inhibition of L-HBs chaperone HSC70 induced degradation of L-HBs with mutant PreS2 via the microautophagy-lysosomal pathway [37]. We consider that YZH-106 binds to the PreS2 domain and might inhibit translocation of L- and M-HBs across the ER membrane topologically, thereby leading to their lysosomal degradation in cytoplasm.

In this study, we observed that YZH-106 had a medium binding affinity of −6.77 kcal/mol to PreS1-PreS2 protein. It will be helpful to optimize the structure of YZH-106 to increase its affinity and specificity to the PreS2 domain for the design of more potent anti-HBV drugs. In addition, we only assessed the inhibitory activity of YZH-106 on HBsAg in vitro and established the proof of principle that rupestonic acid derivatives have the potential for the development of novel compound drugs targeting HBsAg. Experiments with HBV transgenic mice and AAV/HBV-infected mice models are needed to further evaluate the HBsAg-suppressing capability of YZH-106 and its optimized compounds and reversion of T cell tolerance to HBV.

## 5. Conclusions

In summary, we provide evidence that YZH-106 can directly bind to the PreS2 domain of HBV L- and M-HBsAg, thereafter promoting their lysosomal degradation and inhibiting HBV expression and replication. Our findings show a parent compound as a potential anti-HBV agent by novel inhibitory mechanisms. Its structural and functional optimization need to be further explored for the development of novel anti-HBV agents.

## Figures and Tables

**Figure 1 viruses-16-01151-f001:**
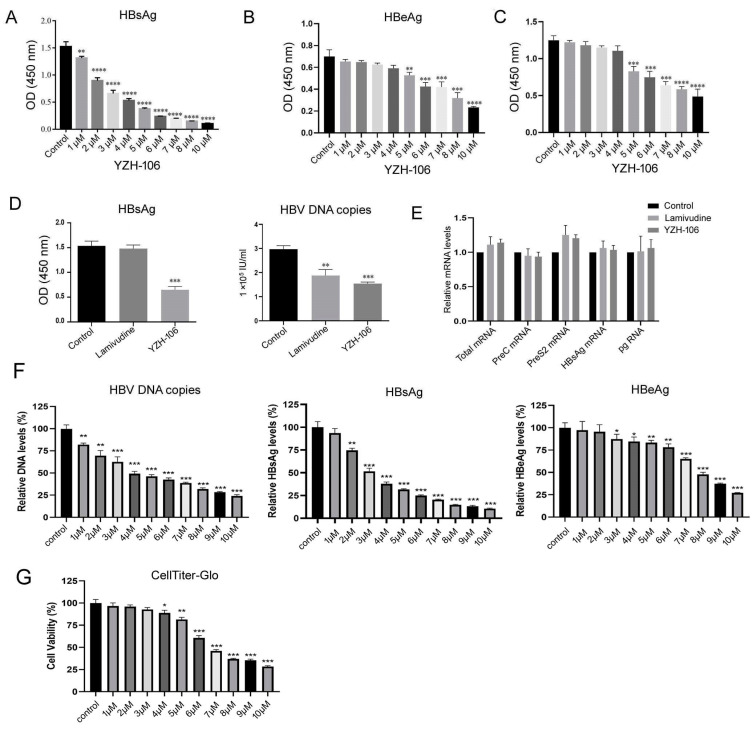
YZH-106 inhibited HBsAg secretion and decreased L-HBsAg and M-HBsAg protein levels. (**A**–**C**) HepG2.2.1.5 cells were treated with the indicated doses of YZH-106 or DMSO as a control. At 72 h after treatment, levels of HBsAg (**A**) and HBeAg (**B**) in the supernatant were measured using ELISA, and cell proliferation was detected by CCK-8 assay (**C**). (**D**,**E**) HepG2.2.1.5 cells were treated with 3 μM YZH-106 or DMSO and 3 μM lamivudine as a control. At 24 h after treatment, HBsAg or HBV DNA copies in the supernatant (**D**) and HBV mRNA levels in cells (**E**) were quantified using ELISA or real-time PCR, respectively. GAPDH was used as an internal control. (**F**) HepG2-NTCP cells infected with HBV were treated with the indicated doses of YZH-106 or DMSO as a control. At 72 h after treatment, levels of HBsAg, HBeAg, and HBV DNA in the supernatant were measured using ELISA or real-time PCR, respectively. (**G**) HepG2-NTCP cells were treated with the indicated doses of YZH-106 or DMSO as a control. At 72 h after treatment, the cellular cytotoxicity was measured using the CellTiter-Glo assay. Data are presented as the means ± SD from three independent experiments. * *p* < 0.05, ** *p* < 0.01, *** *p* < 0.001, and **** *p* < 0.0001 compared to control.

**Figure 2 viruses-16-01151-f002:**
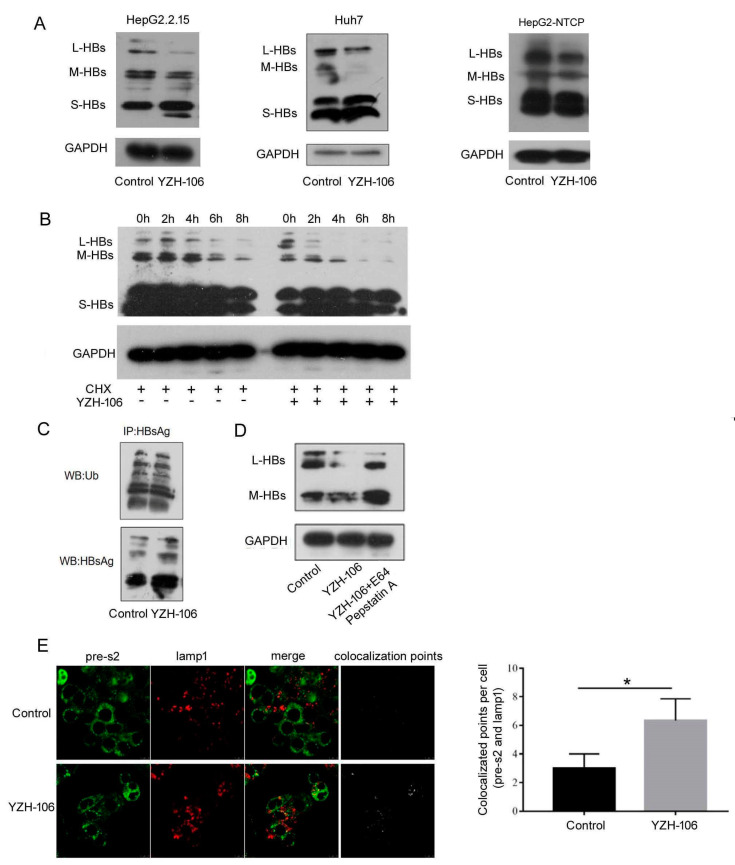
YZH-106 induced lysosomal degradation of L- and M-HBs proteins. (**A**) HepG2.2.1.5 and Huh-7 cells transfected with 1.3 copies of HBV or HepG2-NTCP cells infected with HBV were treated with 3 μM YZH-106 or DMSO as a control, respectively. At 24 h after treatment, L-HBs, M-HBs, and S-HBs protein levels were detected using Western blotting. (**B**) HepG2.2.1.5 cells in the presence of the translation inhibitor CHX were treated with 3 μM YZH-106 or DMSO as a control for the indicated times. Then L-, M-, and S-HBs protein levels were detected using Western blotting. (**C**) HepG2.2.1.5 cells were treated with 3 μM YZH-106 for 24 h and exposed to 20 µM MG132 for 6 h before lysis. HBs protein was immunoprecipitated and subjected to immunoblot with an antibody specific to ubiquitin. (**D**) HepG2.2.1.5 cells were treated with 3 μM YZH-106 alone for 24 h or together with E64d and Pepstatin A. L- and M-HBs proteins were analyzed using Western blotting. (**E**) Immunofluorescence staining was performed to analyze the co-localization of PreS2 (Green) and lamp1 (red) in YZH-106 treated HepG2.2.1.5 cells. The co-localization points were computed using Image J (https://imagej.net/ij/). Data are presented as the means ± SD from three independent experiments. * *p* < 0.05 compared to control. The experiments were performed twice with similar results.

**Figure 3 viruses-16-01151-f003:**
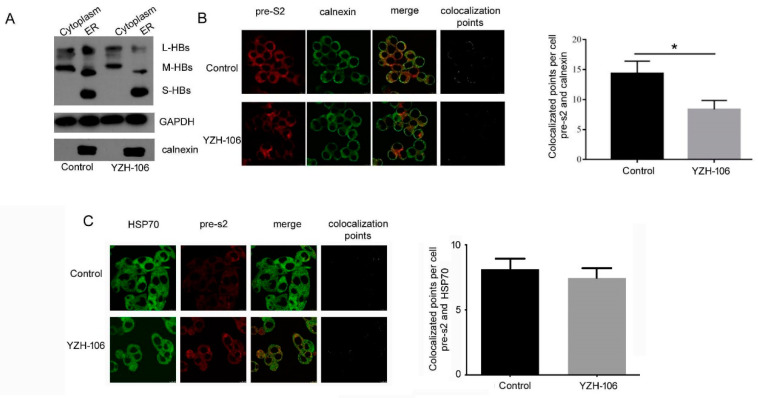
YZH-106 treatment reduced L- and M-HBs protein levels in the ER. (**A**) The cytoplasmic and ER components in HepG2.2.15 cells treated with YZH-106 were isolated, and L-, M-, and S-HBs proteins were examined using Western blotting analysis. (**B**,**C**) Immunofluorescence was performed to analyze the co-localization of PreS2 (red), ER marker Calnexin (green), (**B**) or cytoplasmic marker HSP70 (green) (**C**) in YZH-106 treated HepG2.2.1.5 cells, and the co-localization points were computed using Image J (https://imagej.net/ij/). Data are presented as the means ± SD from three independent experiments. * *p* < 0.05 compared to control.

**Figure 4 viruses-16-01151-f004:**
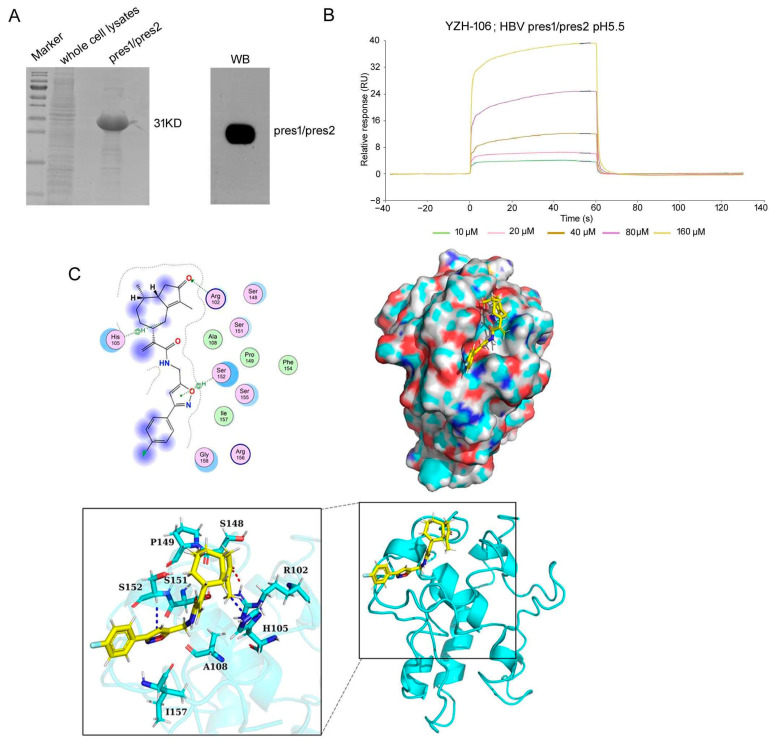
YZH-106 binds to PreS2 domain. (**A**) Expression and purification of His-tagged PreS1-PreS2 fusion protein in *E. coli*. The purified protein was subjected to SDS-PAGE and stained with Coomassie blue or immunoblotted with an anti-PreS2 Ab. (**B**) Interaction of YZH-106 and PreS1-PreS2 fusion protein was evaluated using a direct binding assay using the Biacore system. (**C**) YZH-106 docking model with PreS1-PreS2 fusion protein. Top left: the 2D binding mode of YZH-106 and PreS1-PreS2. Top right: the surface binding mode of YZH-106 and PreS1-PreS2. Bottom: the 3D binding mode of YZH-106 and PreS1-PreS2.YZH-106 and PreS1-PreS2 residues are colored in yellow and cyan, respectively. The residue Arg102 adjacent or within PreS2 forms a hydrogen bond with YZH-106. Additionally, residues His105 and Ser152 each engage in two H-Pi conjugation interactions with YZH-106. The interaction amino acids between YZH-106 and PreS1-PreS2 are shown as blue or red sticks, and non-carbon atoms are colored according to their chemical identity (C, cyan or yellow; O, red; N, blue; H, white).

**Figure 5 viruses-16-01151-f005:**
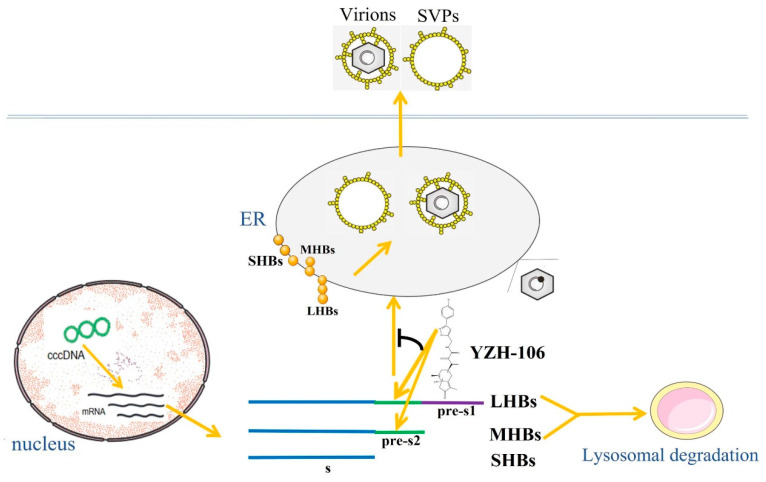
Schematic figure of YZH-106-mediated suppression of HBsAg. HBV L-HBs, M-HBs, and SHBs proteins are translated in the cytoplasm of infected cells. YZH-106 could bind to the PreS2 domain of L-HBs and M-HBs protein, thereby inhibiting their entry to the ER and promoting their lysosomal degradation. As a sequence, the viral envelope assembly in the ER and the secretion of HBsAg was decreased.

**Table 1 viruses-16-01151-t001:** Name and sequence of qRT-PCR primers used in this study.

Name	Sequence 5′-3′	Location in HBV Sequence
pgRNA forward	TCTTGCCTTACTTTTGGAAG	2219–2237 bp
pgRNA reverse	AGTTCTTCTTCTAGGGGACC	2363–2382 bp
total RNA forward	CTCCCCGTCTGTGCCTTCTC	1547–1566 bp
total RNA reverse	TCGGTCGTTGACATTGCTGA	1676–1695 bp
PreC RNA forward	GAGTGTGGATTCGCACTCC	2219–2237 bp
PreC RNA reverse	GAGGCGAGGGAGTTCTTCT	2374–2392 bp
HBsAg RNA forward	CACATCAGGATTCCTAGGACC	168–188 bp
HBsAg RNA reverse	GGTGAGTGATTGGAGGTTG	323–341 bp
PreS2 RNA forward	CCACCATGCAGTGGAACTC	3169–5 bp
PreS2 RNA reverse	TGTGTTCTCCATGTTCGGTG	149–169 bp

**Table 2 viruses-16-01151-t002:** Docking scores.

Ligand	Receptor	Binding Energy (kcal/mol)
Compound	pre-s1	−2.89
Compound	pre-s1/pre-s2	−6.77

## Data Availability

All data are available in the main text or the Appendix A.

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
