# Peer review of "Rupestonic Acid Derivative YZH-106 Promotes Lysosomal Degradation of HBV L- and M-HBsAg via Direct Interaction with PreS2 Domain"

_viruses, 2024, doi:10.3390/v16071151_

Round 1
Reviewer 1 Report
Comments and Suggestions for Authors
The paper entitled "Rupestonic acid derivative YZH-106 promotes lysosomal degradation of HBV L- and M-HBsAg via direct interaction with PreS2 domain" is of interest in the field. The paper is well written, the technical approaches are sound.
The potential of this inhibitor is interesting and requires further studies. The authors did present several cell models, such as stably transfected HepG2 but the results are not presented in the main body of the text. May be the authors could write one or 2 sentences to assess/validate the coherence of the results in the different models. The mechanism by which the Rupestonic acid derivative YZH-106 interact with PRE S2 may be further detailed or at least as hypothesis. Since this derivatives was used for its antiviral activity for flue the authors may refer to this literature as "background".
Author Response
Comments1:The potential of this inhibitor is interesting and requires further studies. The authors did present several cell models, such as stably transfected HepG2 but the results are not presented in the main body of the text. May be the authors could write one or 2 sentences to assess/validate the coherence of the results in the different models.
Response 1: According to your suggestion, we have added the related contents in the Results section. The following sentence has been added on lines 235-236 of page 3. “YZH-106 induced a similar decrease of L- and M-HBs in these two cell models.”
Reviewer 2 Report
Comments and Suggestions for Authors
This is an interesting manuscript characterizing a compound that appears to bind to HBsAg and promote its lysosomal degradation. I think this manuscript should be accepted pending the following revisions:
1. Please determine the antiviral efficacy (EC50) of YZH-106 in HBV-infected HepG2-NTCP cells using HBV DNA, HBsAg, and HBeAg as readouts. Also please determine the cellular cytotoxicity (CC50) in HepG2-NTCP cells using Cell Titer Glo. These results will demonstrate the selectivity for antiviral activity vs. cytotoxicity in HBV-infected cells.
2. Please confirm that both L-HBsAg and M-HBsAg are degraded when HBV-infected HepG2-NTCP cells are treated with YZH-106. For western blotting HBsAg, I recommend the following antibodies - for HBV genotype D: Fitzgerald Industry International #20-HR20. For HBV genotype C or HBV genotype A: ImmunoDiagnostics, cat#111.
Author Response
Comments 1: This is an interesting manuscript characterizing a compound that appears to bind to HBsAg and promote its lysosomal degradation. I think this manuscript should be accepted pending the following revisions: Please determine the antiviral efficacy (EC50) of YZH-106 in HBV-infected HepG2-NTCP cells using HBV DNA, HBsAg, and HBeAg as readouts. Also please determine the cellular cytotoxicity (CC50) in HepG2-NTCP cells using Cell Titer Glo. These results will demonstrate the selectivity for antiviral activity vs. cytotoxicity in HBV-infected cells.
Response 1: According to your suggestions, HepG2-NTCP cells were infected with HBV virions from CHB patients and were treated with the indicated doses of YZH-106 or DMSO as a control. At 72 h after treatment, levels of HBsAg, HBeAg or HBV DNA in the supernatant were measured by ELISA or real-time PCR respectively. The EC50 of YZH-106 on HBV DNA, HBsAg, and HBeAg in HBV-infected HepG2-NTCP cells was determined respectively. Please see Figure 1F. Cell viability was analyzed using the CellTiter-Glo assay (Fig. 1G).
The following sentences have been added on lines 91-98 of page 3, lines 103-114 of page 3, and lines 190-192 of page 5 in the “Material and methods” section.,
“HepG2-hNTCP cells, which stably express a functional receptor for HBV human sodium taurocholate co-transporting polypeptide (hNTCP) in HepG2 cells, were provided by Prof. Wenhui Li (National Institute of Biological Sciences, Beijing). HepG2-hNTCP cells were maintained in hepatocytes maintenance medium (PMM) containing Williams E medium supplemented with 5 μg/ml transferrin, 10 ng/ml EGF, 3 μg/ml insulin, 2 mM L-glutamine, 18 μg/ml hydrocortisone, 40 ng/ml dexamethasone, 5 ng/ml sodium selenite, 2% DMSO, 100 U/ml penicillin, and 100 μg/ml streptomycin. Cells were maintained in a 5% CO2 humidified incubator at 37°C, and the medium was changed every 2-3 days.”;
“2.3. HBV infection assay
HBV genotype B virus was obtained by ultracentrifugation of plasma from three chronic HBV carriers. CHB patients were defined as follows: people who had chronic HBV infection with serum HBsAg positive for ≥6 months and may have exhibited symptoms of hepatitis and abnormal hepatic function. Patient sera were concentrated with Ultra-15 centrifugal filter units containing Ultracel-50 membranes (Millipore Corp., Bedford, MA, USA). After cesium chloride density gradient ultracentrifugation, dialysis against PBS buffer, HBV virions were prepared for infection of HepG2-NTCP cells.
Viral infections were conducted in 12-well plates at multiplicities of genome equivalents of 100. Briefly, 3.2 × 105 epG2-hNTCP cells were inoculated with 3.2 × 107 copies of genome equivalent virus and incubated for 16 h in PMM medium containing 4% PEG-8000. Cells were then washed with medium three times and maintained in PMM medium for 3 d.”;
“2.13 Ethics statement
All human subjects provided written informed consent. The study protocol was approved by the Ethics Committee of the Fifth Medical Centre of PLA General Hospital.";
Also, the following sentences have been added on lines 216-221 of page 6 and lines 221-225 of page 6 in the “Results” section.
“Furthermore, HepG2-NTCP cells infected HBV from CHB patients were treated with YZH-106 and then HBsAg, HBeAg and HBV-DNA in the cell supernatant were detected. As shown in Fig. 1F, YZH-106 induced reduction in HBV DNA copies, HBsAg and HBeAg levels in a dose dependent manner. More decrease of HBsAg than HBeAg was observed after treatment with YZH-106. The EC50 of YZH-106 on HBV-DNA, HBsAg, and HBeAg was around 4 μM, 3μM and 8μM, respectively.”;
“To exclude the possibility that inhibition of HBV by YZH-106 was caused by the cellular cytotoxicity in HepG2-NTCP cells, cell viability was analyzed using the CellTiter-Glo assay. The results demonstrated that YZH-106 at concentrations of 4 μM or less could hardly influence cell viability (Fig. 1G), indicating that YZH-106 exerted antiviral activity not mainly through cytotoxicity.”
Thanks for your comments that greatly strengthened this manuscript.
Comments 2: Please confirm that both L-HBsAg and M-HBsAg are degraded when HBV-infected HepG2-NTCP cells are treated with YZH-106. For western blotting HBsAg, I recommend the following antibodies - for HBV genotype D: Fitzgerald Industry International #20-HR20. For HBV genotype C or HBV genotype A: ImmunoDiagnostics, cat#111.
Response 2: As you suggested, HepG2-NTCP cells infected with HBV were treated with YZH-106 for 24 h. Then L-HBs, M-HBs and S-HBs were detected by western blotting. Please see Figure 2B.The following sentences have been added on lines 232-234 of page 6. “In addition, reduction of L- and M-HBs was also observed in HBV infected- HepG2-NTCP cells under YZH-106 treatment compared to control (Fig. 2A).”
Thanks for your comments.
Round 2
Reviewer 2 Report
Comments and Suggestions for Authors
No further comments